# Feeding-Induced Changes of Bacteriolytic Activity and the Pattern of Bacteriolytic Compounds in the Stomach and Small Intestine of the Haematophagous Bug *Triatoma infestans* (Klug, 1834) (Reduviidae, Triatominae)

Christian K. Meiser, Jennifer K. Pausch and Günter A. Schaub *

Zoology/Parasitology Unit, Faculty of Biology and Biotechnology, Ruhr-University Bochum,
44780 Bochum, Germany; christian.meiser@ruhr-uni-bochum.de (C.K.M.); jennifer_pausch@gmx.de (J.K.P.)
* Correspondence: guenter.schaub@rub.de; Tel.: +49-(234)-3224587

**Abstract:** Intestinal homeostasis mechanisms of the haematophagous triatomines regulate the development of mutualistic symbionts and other gut bacteria. Investigating antimicrobial compounds of these insects, we have determined spectrophotometrically that the bacteriolytic activity is between pH 3 and pH 9 using homogenates of fifth instar *Triatoma infestans* stomachs and small intestines from unfed bugs and up to 50 days after feeding. The activity against Gram-positive *Micrococcus luteus* was strongest at pH 4 and pH 7 and was higher in the stomach than in the small intestine. Symbiotic *Rhodococcus triatomae* were not lysed. Lysis of Gram-negative *Escherichia coli* showed a maximum at pH 7 in the stomach and at pH 5 in the small intestine. Bacteriolytic activity against both *M. luteus* and *E. coli* was reduced 24 h after feeding, then increased, and at 50 days after feeding was strongly reduced. In zymographs, the activity against *M. luteus* was mainly correlated to proteins of about 16 kDa. At different periods of time after feeding, seven bands of lysis appeared between 15 and 40 kDa and more bands using extracts of the small intestine than those of the stomach. This is the first proof for the synthesis of antibacterial proteins of 22–40 kDa in triatomines.

**Keywords:** antibacterial activity; antibacterial compounds; *Escherichia coli*; gut bacteria; *Micrococcus luteus*; *Rhodococcus triatomae*; *Triatoma infestans*; zymogram

## 1. Introduction

Triatomines, vectors of *Trypanosoma cruzi* (Chagas, 1909) (the etiologic agent of Chagas disease in the New World), ingest large amounts of blood, with nymphs taking 6–12 times their own body weight [1]. The blood passes the oesophagus and the beginning of the midgut, the cardia, and is stored in the strongly distensible stomach. There, the blood remains mainly undigested, but is concentrated and the erythrocytes are lysed. Small portions of the concentrated blood are passed from the stomach to the digestive part of the midgut, the small intestine (summarized by [1,2]). Digestion takes place slowly over a long time, lasting 14 days in fully engorged females of *Triatoma infestans* [3], and more than 35 days in fully engorged fifth instar nymphs of this species [4]. In nymphs, the nutrients are needed for the development to the next instar.

Similarly to the majority of blood sucking insects, triatomines require mutualistic symbionts for normal development [5,6]. Aposymbiosis induces strong pathological effects (summarized by [7,8]). So far, only Gram-positive Actinomycetales have been classified as symbiotic (summarized by [6]). The bugs acquire the symbionts via coprophagy, a behaviour that occurs regularly after blood ingestion. Air-borne microorganisms also gain access to the gut by means of air swallowed before moulting in order to increase the volume of the body or by contact of the mouthparts with the skin before and after feeding [9,10]. In 16s ribosomal DNA sequencing in a cultivation-independent method,

the gut bacteria composition varies according to the triatomine species, the developmental stage and the region of the intestinal tract (summarized by [11]). Additionally, with bacterial culture methods, many different microorganisms have been isolated from the gut (summarized by [6,8]). Since the ingestion of a mixture of blood and faeces of triatomines kills *T. infestans* (Schaub unpublished), triatomines must possess intestinal homeostasis mechanisms to inhibit the development of low numbers of non-symbiotic intestinal bacteria after coprophagy. In addition, the population of mutualistic symbionts is regulated as follows: after blood ingestion, the symbiotic bacteria develop strongly but not indefinitely in the cardia and stomach of *T. infestans* and *Rhodnius prolixus* Stål, 1859 but show a significantly lower density in the small intestine [12]. A blood meal-induced proliferation of gut bacteria—without identification of mutualistic symbionts—was also evident in *R. prolixus* and the mosquito *Aedes aegypti* (Linnaeus, 1762) [13,14].

Bacteriolytic activity in the gut of triatomines was first reported by Duncan [15]. In *R. prolixus*, bacteriolytic activity varied according to the feeding status and was much stronger in the stomach than in the small intestine. This activity was attributed to the activity of lysozymes [16]. Recently, in-gel zymograms were used to investigate intestinal antibacterial compounds of *Triatoma brasiliensis* Neiva, 1911. A lysozyme from *T. brasiliensis*, synthesized in *Escherichia coli*, showed muramidase activity against *M. luteus* and activity under alkaline and acidic conditions, indicating a digestive function in different compartments of the insect's midgut [17]. The lysozymes of triatomines so far described belong to the chicken-type (c-type) lysozymes, possessing a molecular mass of about 15 kDa and a neutral to basic pI [9,18,19]. In another lysozyme from the digestive tract of *T. infestans,* the conserved amino acid residues at the active site are replaced [20]. Such a lysozyme has not been detected so far in *R. prolixus* [19,21].

According to molecular biological data and transcriptomes, not only genes encoding lysozymes (15 kDA), but also those encoding other antimicrobial peptides are expressed in the intestine of *R. prolixus*, *T. infestans* and *T. brasiliensis* and these include the following: defensins (4 kDa), pacifastin-like protease inhibitors (4 kDa), prolixicin (11 kDa), the Kazal-type inhibitor RpTI (11.5 kDA), TiAp (12.5 kDa) and attacins (20–23 kDa) [22–27]. Whereas RpTI of *R. prolixus* is a double domain protein [24], in the triatomine *Panstrongylus megistus* (Burmeister, 1835) the first domain of the Kazal-type inhibitor (7 kDa) is suggested to act as an antimicrobial peptide [28]. In addition, the antimicrobial compounds diptericin (9 kDa), triatox (14.8 kDa), and trialysin (22 kDA) are synthesized in the salivary glands of triatomines [26]. Other low-molecular mass compounds, involved in antibacterial responses, e.g., nitric oxide, are also present in the intestinal tract of triatomines [29]. Besides these low-molecular mass antibacterial compounds and bacteriolytic peptides, compounds with bacteriostatic function are also present in the intestine of insects [30,31], but have not been described for triatomines.

Finally, since the feeding-induced time course of bacteriolytic activity has only been considered for the small intestine of *T. brasiliensis* [17], the present investigation focussed on the time course of this activity in the stomach and small intestine of *T. infestans*. Turbidity assays and zymograms with three species of bacteria at different pH values indicate the synthesis of different bacteriolytic compounds at different times after feeding in both regions of the midgut of triatomines.

## 2. Results

### 2.1. Weight of Stomach and Small Intestine and Concentration of Soluble Proteins

The weight of the stomach was strongly increased by the ingested blood from about 30 mg in unfed fifth instar nymphs to about 210 mg at 1 day after feeding (daf) (Figure 1, white column). Then, it decreased until 40 daf to double the weight of the stomach of unfed insects, reaching at 50 daf half of the initial weight. At all times after feeding, means differed significantly from those of unfed insects ($p < 0.05$). The concentration of soluble proteins per mg stomach varied considerably in unfed fifth instar nymphs (Figure 1, grey column), showing no statistically significant difference to the mean concentration at 1 daf

($p > 0.05$). It remained at a similar level until 40 daf, and at 50 daf it was strongly and significantly reduced compared to the previous concentration ($p < 0.05$).

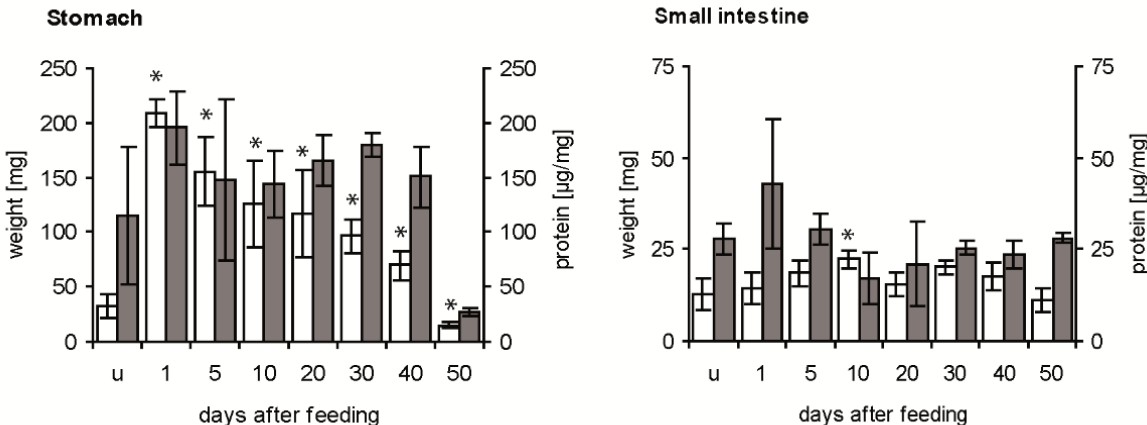

**Figure 1.** Time course of weight of stomach and small intestine (white column) and concentration of soluble proteins (grey column) in the digestive tract after a blood meal. Data represent means and standard deviations of measurements using three independent pools, each containing the stomachs or the small intestines of three *T. infestans*, unfed (u) and at different days after feeding (1, 5, 10, 20, 30, 40 and 50). Significant differences to the data of unfed bugs are indicated by an asterisk ($p \leq 0.05$).

Compared with the weight of unfed nymphs, the weight of the small intestine was increased significantly at 10 daf ($p < 0.05$), remained at this level until 40 daf and then decreased significantly below the weight at 10 daf ($p < 0.05$; comparison not indicated in Figure 1) (Figure 1, white column). The concentration of soluble proteins in unfed insects, dissected 40 daf of fourth instar nymphs, varied strongly at 1 daf and was then similar with a concentration of about 25 $\mu$g mg$^{-1}$ at 50 daf (Figure 1, grey column). Comparing stomach and small intestine, in the first 40 daf the stomach was always 3 to 14-fold heavier than the small intestine, while at 50 daf they had similar weights. Except for the similar concentrations of soluble protein at 50 daf, concentrations in the stomach were 4–8 times higher than in the small intestine.

*2.2. Bacteriolytic Activity*

Regarding the comparison of the pH dependency of the antibacterial activities of the stomach homogenate supernatants of fifth instar nymphs at 20 daf, the activity against *Micrococcus luteus* was highest at pH 4, strongly reduced at pH 5 and increased above pH 5 to a broad range of high activity between pH 6 and pH 9 (Figure 2, filled triangle, solid line). The activity of the small intestine supernatants showed a similar profile (Figure 2, open triangle, solid line). Using the homogenates from bugs dissected at different daf, the activity peaks were always at pH 4 and pH 7. The optical densities of the suspensions with the mutualistic symbiont of *T. infestans*, *Rhodococcus triatomae*, did not change during the incubations using supernatants from both parts of the intestine and from bugs dissected at different daf (data not shown). Using *Escherichia coli*, the optical densities increased during the incubations at pH 3 and pH 4 (data not shown). The supernatants from the stomach possessed the highest activity against this Gram-negative bacterium at pH 7 to 9 (Figure 2, filled triangle, dotted line), while those from the small intestine were the highest at pH 5 to 7 (Figure 2, open triangle, dotted line). Using the homogenates from bugs dissected at different daf, maxima were similar at pH 7 and pH 5 (data not shown).

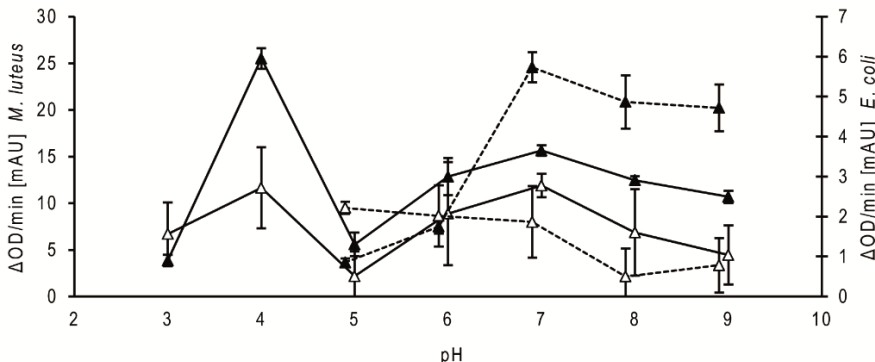

**Figure 2.** Bacteriolytic activity against *M. luteus* and *E. coli* at different pH. Data represent means and standard deviations of the bacteriolytic activity against *M. luteus* (solid lines) and *E. coli* (dotted lines) by extracts of the stomach (filled triangles) and small intestine (open triangles) dissected at 20 daf calculated from changes in optical density measured for 15 min using 3 independent pools, each containing the stomachs or the small intestines of three nymphs of *T. infestans*.

In the determinations of the time course of bacteriolytic activity after feeding at the two pH levels giving maximal activity (see previous experiment), the newly ingested blood (1 daf) reduced the bacteriolytic activity of the stomach extracts against *M. luteus* by about 50% at pH 4 and pH 7 ($p < 0.01$; $p < 0.05$) (Figure 3, grey column). From that time point onwards, the bacteriolytic activity increased until 20 and 40 daf at pH 4 and pH 7, respectively, and then decreased significantly between 40 and 50 daf ($p < 0.01$, $p < 0.001$). In the small intestine, the activity against *M. luteus* at pH 4 was much lower than in the stomach, showed a retarded time dependency at 5 daf, then increased and remained constant until 50 daf (Figure 3, white column). Additionally, at pH 7 the activity was lower than in the stomach, possessing a similar time course to pH 4, but with a much stronger significant reduction at 50 daf. In statistical comparisons, only the activities in fed bugs at 20 and 30 daf and at 40 and 50 daf were significantly different ($p < 0.05$, $p < 0.001$).

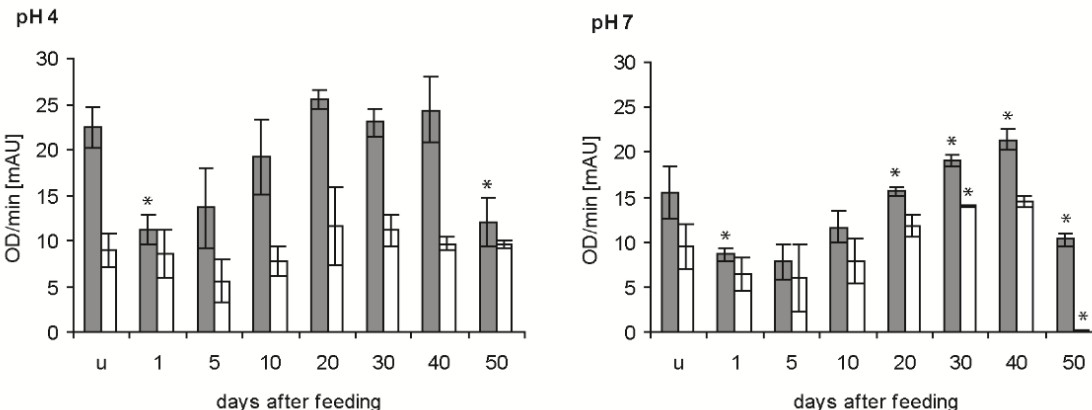

**Figure 3.** Time course of the bacteriolytic activity against *M. luteus* by extracts of the stomach (grey column) and small intestine (white column) of *T. infestans,* unfed (u) and at different days after feeding (1, 5, 10, 20, 30, 40 and 50), at pH 4 and pH 7. Data represent means and standard deviations of activity calculated from changes in optical density measured every minute over 15 min using three independent pools, each containing the extracts of stomachs or small intestines of three bugs. An asterisk above a column indicates a statistically significant difference to the mean activity of the previous sample ($p$ at least $\leq 0.05$).

Bacteriolytic activity against *E. coli* was significantly increased in the stomach at pH 5 after blood ingestion ($p < 0.05$), decreased to a minimum at 40 daf and increased at 50 daf ($p < 0.05$) (Figure 4, grey column). At pH 7, the activity in the stomach was significantly decreased at 5 daf compared with unfed insects ($p < 0.05$), then increased strongly to 20 daf.

This activity remained stable until 40 daf and was significantly reduced at 50 daf ($p < 0.01$). In the small intestine of unfed nymphs and until 10 daf, the activity at pH 5 was very similar to that in the stomach, higher between 20 and 40 daf and 50% lower at 50 daf (Figure 4, white column). At pH 7, the activity was always significantly lower in the small intestine than in the stomach, but the time course was similar, showing a maximum at 20 and 30 daf and a significant decrease at 40 and 50 daf ($p < 0.01$).

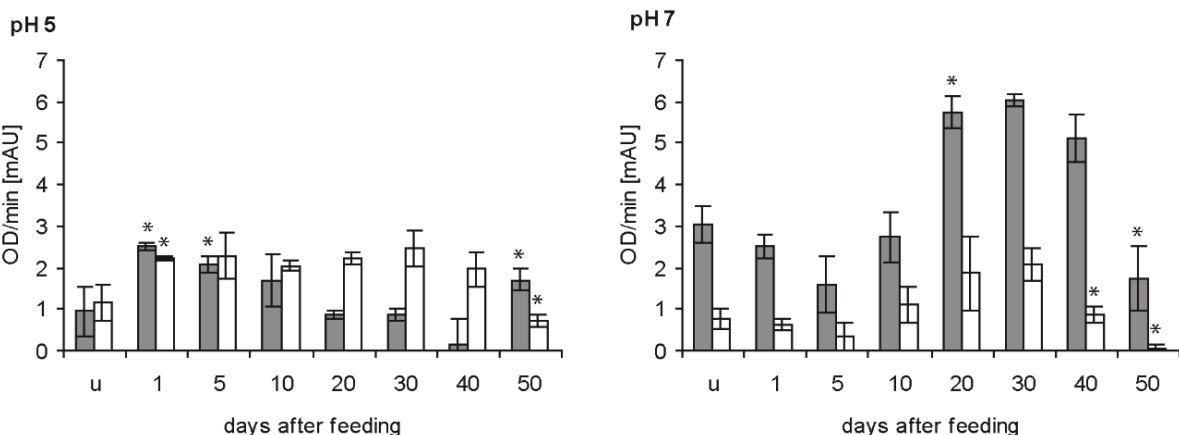

**Figure 4.** Time course of the maximal bacteriolytic activity against *E. coli* by extracts of the stomach (grey column) and small intestine (white column) of *T. infestans*, unfed (u) and at different days after feeding (1, 5, 10, 20, 30, 40 and 50), at pH 5 and pH 7. Data represent mean and standard deviations of activity calculated from changes in optical density measured every minute over for 15 min using three independent pools, each containing the extracts of stomachs or small intestines of three bugs. An asterisk above a column indicates a statistically significant difference to the mean activity of the previous sample ($p$ at least $\leq 0.05$).

### 2.3. Protein Banding after Electrophoresis and Bacterial Lysis in Gels

After electrophoresis of the supernatant of the stomach homogenates of unfed nymphs and of bugs dissected at different times after feeding, the majority of proteins were located at 14, 28 and 57 kDa, about 65% of total protein content of the respective sample at 40 daf (Figure 5). At 50 daf the samples contained only traces of proteins. In the supernatants of the homogenates of the small intestine, the pattern of protein bands was more complex than in the stomach. The concentration of proteins of 14 kDa increased strongly after feeding and was similar until 20 daf, about 40% of total protein at 1 to 30 daf. These proteins were no longer detectable at 40 and 50 daf.

After zymography of the stomach homogenates using *E. coli* as substrate, no lysis regions were found (data not shown). Using *M. luteus* as substrate and incubations at pH 4 or pH 7, lysis bands appeared at about 16, 22 and 40 kDa (Figure 6). At pH 4, the activity at 16 kDa was present in unfed nymphs, nearly lacking in the first 5 daf, strong at 20–40 daf, and strongly reduced at 50 daf. The activity at about 22 kDa was much lower than the activity at 16 kDa, but showed a similar time course. At 40 kDa, the activity was first visible at 20 daf and then became much lower at 30 and 40 daf. After an incubation at pH 7, the lysis pattern was similar, showing lytic activity in unfed insects at 16 kDa, much lower activities at 1 and 5 daf and an increase afterwards until 40 daf. An additional weak lytic activity occurred at 22 kDa in unfed insects and at 30 and 40 daf. At 40 kDa the activity was detectable at 20 to 40 daf, with the highest activity at 20 daf. Whereas the lysis activity at 16 kDa was comparable at both pH levels, the activities above 16 kDa were stronger after incubations at the acidic pH (Figure 6).

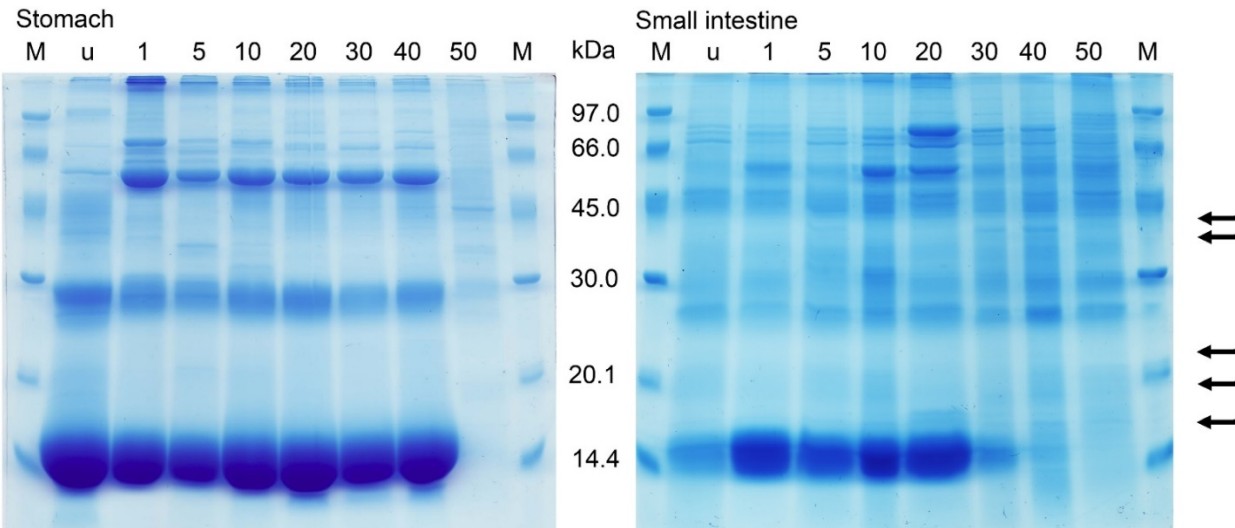

**Figure 5.** Protein profiles in non-reducing SDS-PAGE of the soluble extracts of 0.5 mg stomach and 1 mg small intestine of *T. infestans*, unfed (u) and at different days after feeding (1, 5, 10, 20, 30, 40 and 50). M: marker proteins. The figure represents one typical gel out of three separations. Arrows indicate lysis zones from stomach and small intestine extracts according to Figures 6 and 7.

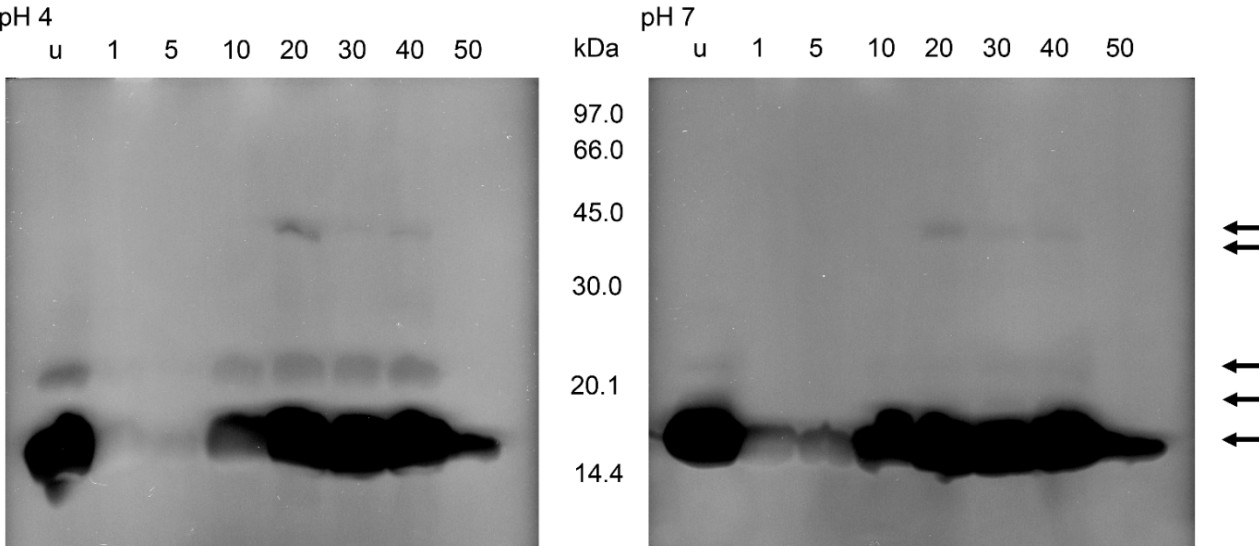

**Figure 6.** Bacteriolytic activity in non-reducing SDS-PAGE of the soluble extracts of the stomach of *T. infestans*, unfed (u) and at different days after feeding (1, 5, 10, 20, 30, 40 and 50) after an incubation overnight at pH 4 and pH 7 using *M. luteus* as a substrate. The figure represents one typical zymogram out of three separations. The gel was processed in a black–white conversion to enable the recognition of weak regions of lysis. Arrows indicate lysis zones from stomach and small intestine extracts.

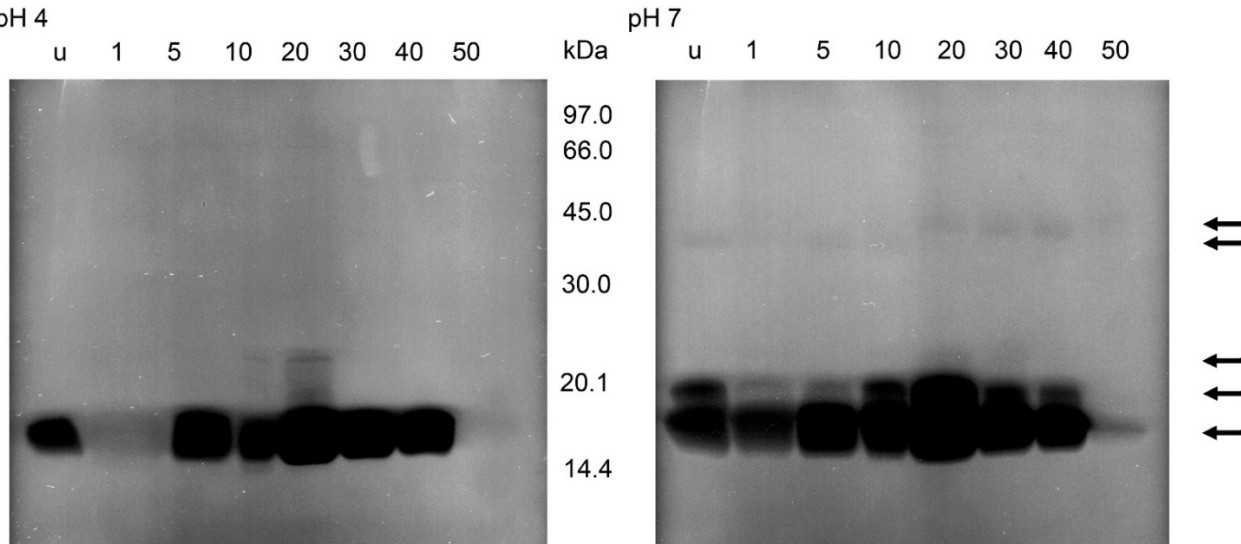

**Figure 7.** Bacteriolytic activity in non-reducing SDS-PAGE of the soluble extracts of the small intestine of *T. infestans*, unfed (u) and at different days after feeding (1, 5, 10, 20, 30, 40 and 50) after an incubation overnight at pH 4 and pH 7 using *M. luteus* as a substrate. The figure represents one typical zymogram out of 3 separations. The gel was processed in a black–white conversion to enable the recognition of weak regions of lysis. Arrows indicate lysis zones from stomach and small intestine extracts.

After the incubations of the gels containing the separations of the supernatants of small intestine homogenates and *M. luteus* at pH 7, the zymograms showed bacteriolytic activities at 16, 19, 36 and 40 kDa (Figure 7). The lysis at 16 kDa was stronger than in the other regions and present at all times after feeding with a maximum at 20 daf and a strong reduction at 50 daf. Lysis activity at 19 kDa occurred in unfed insects and until 40 daf. Activity at the higher molecular masses changed between 10 and 20 daf; until 20 daf a very faint band was detectable at 36 kDa then at 40 kDa. In incubations at pH 4, activity at 16 kDa was strongly reduced at 1 daf and detectable until 40 daf; activity at 22 kDa was present at 10 daf and 20 daf, being strongest at 20 daf. At 10 daf a very weak additional activity was present between 16 and 22 kDa, clearly recognizable in the figure only at 20 daf as a ladder of bands (Figure 7).

In all zymograms, activity against *M. luteus* was never detectable in the molecular range <14 kDa. This was also the case using electrophoresis conditions optimal for the electrophoresis of small peptides, non-reducing conditions and an incubation at pH 5. However, the lysis area of stomach homogenates at about 16 kDa showed two zones with molecular masses of 15 and 16 kDa. The homogenates of the small intestine yielded three independent lysis bands of 15, 16 and 16.5 kDa. In both samples, the activity was highest at 15 kDa (data not shown).

At all lysis bands, no protein bands were evident in the electrophoresis. In addition, in a comparison of stomach and small intestine less bands of lysis were induced by extracts of the stomach that possesses a higher antibacterial activity.

### 3. Discussion

In previous comparisons of feeding-induced changes of bacteriolytic activities in the intestine of triatomines, the regions were homogenized in identical volumes of saline [16,17]. Thereby, the measurements indicated an initial increase in concentrations of protein in the stomach, then a slow decrease until 6 daf and thereafter a rapid decrease [16]. In the present investigation, the volume of saline was adjusted to the weight of the region. The relatively similar concentrations of protein in the stomach of *T. infestans* between 5 and 40 daf more accurately reflects the specific digestive behaviour of triatomines than

the data for *R. prolixus*, since in triatomines no digestion of haemoglobin occurs in the stomach and the concentrated blood is passed in small portions to the small intestine (summarized by [1,11]).

This storage of the nearly undigested blood for a long time in the stomach necessitates an antibacterial defence to prevent the growth of harmful microorganisms. In the first detailed investigation of the bacteriolytic activity in the gut of triatomines, the Gram-positive bacterium, *M. luteus*, was used as the substrate with a sodium acetate buffer ranging from pH 5.0 to 7.5 [16]. The highest bacteriolytic activity of the stomach and small intestine homogenates of *R. prolixus* occurred between pH 5.0 and pH 6.0, marginally peaking at pH 5.5. In the present investigation of *T. infestans*, a universal buffer system was used enabling measurements between pH 3 and pH 9 with a constant ionic strength. Using *M. luteus*, a previously unreported peak of activity was evident at pH 4 with a second occurring at pH 7. The lower pH value has not been measured in the intestine using microelectrodes [32] but seems relevant due to an acidification gradient presumably existing from the wall of the intestine to the lumen, with the latter the position of the microelectrodes. In pH measurements in unfed nymphs of *T. infestans*, the pH values in the stomach and small intestines were approximately 6.3 and 6.6, respectively. After feeding, the values decreased in both regions to a minimum of pH 5.2 at 8 daf and then increased steadily to pH 5.9 at 20 daf [32].

Although the population density of the Gram-positive mutualistic symbiont *R. triatomae* is strongly reduced in the small intestine of *T. infestans* [12], no lysis occurred using homogenates of both gut regions, buffers at pH 3 to pH 9 and the lyophilized symbionts as a substrate. Perhaps the use of viable symbionts during the two growth phases (solitary and mycelial stage) might produce contrasting results.

Gram-negative bacteria were mainly investigated in Hemiptera in studies of humoral immunity in the haemolymph [33–35]. Comparing the intestinal antibacterial activity against Gram-negative or Gram-positive bacteria, stomach contents of the fifth instars of *R. prolixus* had a significantly higher antibacterial activity than contents of the small intestine [36]. The activity against *E. coli* and *Staphylococcus aureus* increased after feeding to a maximum at 7 daf and then decreased. Comparing the time course in *T. infestans* at a pH value similar to that used by Vieira et al. [36], feeding induced a decrease in the activity against *E. coli* in the stomach up to 5 daf. Then, the activity increased strongly to 20 daf and remained stable until 40 daf. The time course of activity in extracts of the small intestine was similar, but the activity always lower. Additionally, the activity against *M. luteus* peaked at 40 daf and showed the same difference between both regions. The differences in the feeding dependency of the bacteriolytic activity between *R. prolixus* [16,36] and *T. infestans* may be caused by the different dilution of the homogenates.

The strongly reduced activities at 50 daf (also against Gram-negative bacteria), agree with a similar phenomenon reported with the haemolymph of immunized *R. prolixus* [34]. Thus, prolonged starvation correlates with a reduction in the antibacterial activity. Only analysis of changes in numbers of bacteria or in starvation-based reduced synthesis or degradation of antimicrobial compounds can clarify the responsible factor(s).

Since the time course of the bacteriolytic activity after feeding in the two gut regions and the pH optima in the activities against different substrates differ, several region-specific intestinal antibacterial compounds are probably involved. The significant reduction in the antibacterial activity after an incubation with trypsin or boiling [36] indicates the importance of peptides/proteins. To specify these compounds, proteins of the homogenates were separated electrophoretically and the gels used for zymography. Recently, this has been studied using *M. luteus* and an incubation at one pH value, pH 5.5, for 3 h [17]. These workers showed with homogenates of the stomach of unfed *T. brasiliensis* that bands of lysis appeared at 14, 16 and 18 kDa. The material of the small intestine of unfed nymphs of *T. brasiliensis* and of nymphs dissected at 3, 5, 10 and 15 daf always induced a strong lysis at 14 kDa, a weak one at 16 kDa—except 3 daf—and at 12 kDa from the fifth daf onwards. The lysis band at 14 kDa was also evident using recombinant lysozyme of *T. brasiliensis* [17].

In the present investigation, after overnight incubation with *M. luteus* at pH 4 or pH 7, the stomach homogenate induced lysis bands at 15, 16, 22 and 40 kDa. The small intestine of *T. infestans* resulted in a similar time course of activity as the stomach and in lysis bands at 15, 16, 16.5, 19, 22, 36 and 40 kDa. The differences to *T. brasiliensis* seems to be caused by the electrophoresis conditions and the longer period of incubation. In *T. brasiliensis* and in *T. infestans* no prominent bands were present in electrophoresis at the positions of lysis bands in zymographs, indicating the low concentrations of the antimicrobial compounds. In the present investigation, for the first time the presence of bacteriolytic proteins of >18 kDa was demonstrated, and a reduction in bacteriolytic activities after prolonged starvation was correlated to a decrease in the concentration of proteins at 16 kDa.

In an attempt to correlate the lysis bands with known antimicrobial peptides, no bacteriolytic peptide of <14 kDa was detected although the expression level of genes encoding for defensins (4 kDa) is increased after blood ingestion [18,37]. Since intestinal defensins of the stable fly *Stomoxys calcitrans* (Linnaeus, 1758) are bound intracellularly in SDS-stable complexes to a serine protease and are released into the lumen of the gut [38], one of the lysis zones in the higher molecular range of *T. infestans* homogenates might be such an undissociated complex of defensin with another protein. In addition, in the stomach and small intestine, it could either be present as an inactive proform or not detectable as active due to the electrophoresis conditions. This might also be relevant for the pacifastin-like peptide (4 kDa), diptericin (9 kDa) and prolixicin (11 kDa). Whereas the pacifastin-like peptide is present in the gut, diptericin has only been found in the salivary glands, and genes of prolixicin are expressed in the intestinal wall [22,26,36,39]. In an in vitro assay, the growth of the mutualistic Gram-positive symbiont of *R. prolixus*, *Rhodococcus rhodnii*, was not affected by recombinant prolixicin. Prolixicin is very active against Gram-negative and less active against Gram-positive bacteria [38]. Three other antimicrobial peptides in this range of molecular masses, PmStKAZ I, RpTI and TiAP with 6.6, 11.5 and 12.5 kDA, respectively [24,25,28], were also not detected in the zymographs. This was also the case using electrophoresis conditions optimal for the electrophoresis of small peptides.

Considering the zymographs of [17], in *T. infestans*, the main lysis zone at 16 kDa seems to be caused by lysozymes. Besides the molecular mass, the time course of the gene expression also supports this conclusion. The concentration of mRNA encoding lysozyme increased after feeding in the stomach of *T. infestans* and *T. brasiliensis* about 20-fold or 5-fold, respectively [9,18]. In the intestinal tract of *T. infestans*, genes of two lysozymes are expressed and of an additional one in the salivary glands [9,20] (Meiser, unpublished), perhaps inducing adjacent lysis bands. Polymerization of lysozymes has been found [40,41], but the determined molecular masses in the zymograms, which are not n-fold the molecular mass of lysozyme [42], do not support this possibility here.

Beside lysozymes, the lysis band at 15 kDa could be induced by triatox (14.8 kDa), so far only described for the salivary glands [26]). Since attacins (22–23 kDa) are active against Gram-negative bacteria, lysis bands at 22 kDa might result from trialysin (22 kDa), but this is only known for the salivary glands. In the present paper, zymography using the Gram-negative *E. coli* yielded in no lysis bands. Therefore, this method has to be adapted. However, the spectrophotometrical determinations of the bacteriolysis of *E. coli* verified the presence of antimicrobial peptides active against such bacteria. The identity of the other antimicrobial proteins between 16 and 22 kDa and of 19, 36 and 40 kDa remains to be clarified. To our knowledge, no other factors inducing lysis of *M. luteus* and with corresponding molecular masses have been described. Additionally, a combination of two proteins is possible. In larvae of the housefly *Musca domestica* Linnaeus, 1758, bacteria are lysed at a low pH by a combined action of lysozyme and a cathepsin D-like proteinase [43,44]. This might also be relevant in triatomines, since the infection with *T. cruzi* not only increases the expression of the genes of several antimicrobial peptides but also of cathepsin D and also the activity of cathepsin D [45,46], and the induction of antibacterial activity reduces the number of intestinal bacteria (summarized by [11,47]).

Combining the present data on the bacteriolytic activities and the development of mutualistic symbionts in the intestine of triatomines, the new data correlate with peculiarities in the initial development of the symbionts. The symbiotic *R. rhodnii* develops in the stomach of *R. prolixus* from a coccoid form to a branched form, reverting to the coccoid form in the first 5 to 8 daf [48,49]. Thus, during the period of reduced bacteriolytic activities branched forms seem to occur and the reversal to coccoid forms during the subsequent period of increased activities. The initial increase in symbionts [12] correlates with the reduced bacteriolytic activity in *T. infestans*. Since the small intestine possesses a lower bacteriolytic activity than the stomach, the strong reduction in *R. triatomae* by about 99% [12] seems not to be a direct bacteriolytic process by lysozyme, so that other bacteriolytic factors may act in the small intestine.

## 4. Materials and Methods

### 4.1. Insect Origin, Maintenance and Sample Preparation

*T. infestans*, originating from Cochabamba, Bolivia, were reared five generations at $27 \pm 1$ °C and 70–80% relative humidity (RH) and a 16/8 h light/dark cycle. First instar nymphs were fed on mice, all older stages regularly on hens [50]. Groups of about 150 nymphs were maintained in 2 litre beakers. An optimal development was achieved by supplying them with in vitro cultures of the symbiotic *R. triatomae*, which originates from a domestic population of *T. infestans* from Cochabamba, Bolivia [6,51]. Stomachs and small intestines were dissected from unfed fifth instar nymphs 40 daf of fourth instar nymphs. Another batch of unfed fifth instar nymphs were fed and used at 1, 5, 10, 20, 30, 40 and 50 daf. At the last three dates, the nymphs had moulted to the adult stage. The respective stomachs and intestines of three bugs were pooled and weighed before adding 10 μL sterile physiological saline per mg tissue plus contents. After homogenisation with a pestle and centrifugation at $2500 \times g$ for 10 min at 4 °C, the supernatants were centrifuged at $16,000 \times g$ for 15 min at 4 °C. The resultant supernatants were stored at −80 °C. The concentration of soluble proteins was determined by a commercial Bradford assay (Carl Roth, Karlsruhe, Germany), modified according to Zor and Selinger [52].

### 4.2. Determination of Bacteriolytic Activity

Heat inactivated lyophilised *M. luteus* and *E. coli* were obtained from Sigma (Taufkirchen, Germany). *R. triatomae* were cultivated in Standard I Medium (Merck, Darmstadt, Germany). In the exponential growth phase, symbiont cultures were heated (1 h at 120 °C), centrifuged at $3000 \times g$ for 5 min, washed 2 times in 0.9% NaCl and dried.

Bacteriolytic activity was determined as a change in the optical density of a bacterial suspension [53]. Briefly, 10 μL supernatants of homogenates of stomach or small intestine and 90 μL 111 mM Britton and Robinson buffer (a mixture of acetic acid, phosphoric acid and boric acid) adjusted to pH 3, 4, 5, 6, 7, 8 or 9 with 1 M NaOH [54], were added to the bacteria (final concentration 0.015% ($w/v$)). Optical densities, i.e., light scattering, at 450 nm were recorded at 37 °C in 60 s intervals for 15 min in a Model 680 ELISA-reader (Bio-Rad, München, Germany). Increases in optical densities were suggested to be caused by methodological problems, perhaps by an aggregation of bacteria. Reaction velocity was determined as a linear calculation over the whole monitoring period by the Microplate Manager software version 5.2 (Bio-Rad, München, Germany) and corrected by subtracting the value of the blank at the respective pH. Three independent samples were used in duplicate measurements. All statistics were done using Student's *t*-test in pairwise comparisons (Statistica 9.0, Statsoft Europe, Hamburg, Germany).

### 4.3. Electrophoresis and Zymography

For a better comparison, supernatants equivalent to 0.5 mg of stomach or 1 mg small intestine were loaded in each lane. Initially, also lower protein contents were tested. Proteins > 10 kDa were separated according to Laemmli [55] using 13% T separating gels under non-reducing conditions. For a better separation of proteins in the range of 1–10 kDa, 16% T

separating gels were topped with 5 mm of a 10% T gel, and electrophoresis was performed according to Schägger and von Jagow [56]. All separating gels of $75 \times 80 \times 0.5$ mm$^3$ were run in a Hoefer SE 260 (GE Healthcare, Freiburg, Germany) with a maximum voltage of 300 V until the front of the colour marker had left the gel. After electrophoresis, proteins were fixed in a mixture of 40% methanol and 2% phosphoric acid, washed in water 3 times for 20 min, then equilibrated in a mixture of 10% methanol and 2% phosphoric acid and stained overnight in 40% methanol, 2% phosphoric acid, 6% ammonium sulphate and 0.1% Coomassie G250. Gels were destained in water until the background was clear [57].

For zymography, separating gels contained 0.3% lyophilised *E. coli* or *M. luteus* [58,59]. After electrophoresis, proteins were renatured by two incubations for 15 min in 0.1% Triton X-100 in 100 mM Britton and Robinson buffer (either pH 4 or pH 7) [54]. After 15 min in the respective buffer, to wash out Triton X-100, the pH of the buffer solution was verified and the buffer was changed again. Gels were incubated at 37 °C overnight, stained for 1–2 min in freshly prepared 0.1% methylene blue in 0.001% NaOH and destained in water until the stacking gel was clear. Gels were analysed and molecular masses were calculated in reference to a mix of standard proteins (GE Healthcare, Freiburg, Germany) using ImageMaster$^{TM}$ 1D Version 4.0 (GE Healthcare, Freiburg, Germany).

## 5. Conclusions

By analysing the bacteriolytic activity in the stomach and small intestine of the haematophagous bug *T. infestans* between pH 4 and pH 7, the maximal activities against *M. luteus* and *E. coli* at two pH values indicated the presence of different antimicrobial compounds. Considering the extracts of both intestinal regions dissected unfed and at different time points after feeding the activity is affected by feeding and long-term starvation. These changes correlate with changes in zymograms. Lysis at about 16 kDa seems to be induced by lysozymes, but compounds of higher molecular masses need an identification.

**Author Contributions:** Conceptualization: C.K.M.; methodology: C.K.M.; investigation: C.K.M. and J.K.P.; formal analysis: C.K.M. and J.K.P.; writing—original draft prepared: C.K.M. and G.A.S.; supervision: G.A.S.; funding acquisition: G.A.S. All authors have read and agreed to the published version of the manuscript.

**Funding:** The support of the Deutsche Forschungsgemeinschaft (project SCHA 339/17-1) is gratefully acknowledged.

**Institutional Review Board Statement:** The experiments were performed in accordance with German animal welfare registration (No. 23.8720, 20.A.10).

**Informed Consent Statement:** Not applicable.

**Data Availability Statement:** Data are contained within the article.

**Acknowledgments:** We thank Norman A. Ratcliffe for correcting the English style and Nora Medrano Mercado for kindly providing the *Triatoma infestans* strain.

**Conflicts of Interest:** The authors declare no conflict of interest.

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
