# Peer review of "Feeding-Induced Changes of Bacteriolytic Activity and the Pattern of Bacteriolytic Compounds in the Stomach and Small Intestine of the Haematophagous Bug Triatoma infestans (Klug, 1834) (Reduviidae, Triatominae)"

_parasitologia, doi:10.3390/parasitologia2010002_

Round 1

Reviewer 1 Report

Please see the annexed file.

Author Response

Reviewer 1:

Line 30: Deletion of “and”.

I apologize for the mistake and deleted “until” and not “and”. 

Line 79: I would not describe nitric oxide as an antibacterial compound. It is obviously involved in antibacterial responses, but it works more as a signalling molecule. Please rephrase this sentence.

            Done

Line 147: Is this 5 daf? Please clarify.

            I apologize for the mistake and corrected to “5 daf”.

Line 279: Full stop missing

            Done

Line 280/281: Replacement of “seems to” by “may”.

            Done

Reviewer 2 Report

This is a well written report of what I assume to be painstaking work.

Does Micrococcus luteus lyse blood?

Any thoughts as to why the blood is held in the stomach for so long before being processed to "food" or is providing all the nutrients needed for the next stage of nymph or adult?

Author Response

Reviewer 2:

Does Micrococcus luteus lyse blood?

            I found no publication considering this topic.

Any thoughts as to why the blood is held in the stomach for so long before being processed to "food" or is providing all the nutrients needed for the next stage of nymph or adult?

            I included: In nymphs, the nutrients are needed for the development to the next instar. (Lines 36/37)

Reviewer 3 Report

Congratulations on your excellent contribution to the appearance of the microbiota of an important vector species. I made small suggestions regarding nomenclature.

Author Response

Reviewer 3:

Insert descriptor and year whenever bringing the scientific name for the first time.

Done in lines 4, 28, 54, 57 and 62 and also 75/76, 313 and 347.

Insert (Klug, 1834)

            Done in line 4/5.

The correct sequence in this case is: (Reduviidae, Triatominae).

            Done in line 5.

Change Trypanosoma cruzi to Trypanosoma cruzi (Chagas, 1909).

                        Done in line 28.

Insert Stål, 1859

                        Done in line 54.

Insert (Linnaeus, 1762)

                        Done in line 57.

Insert Neiva, 1911

                        Done in line 62.

I included additional corrections in lines 21, 59, 282, 380, 384, 446, 485 and 492-493.